# Input-Cell Attention Reduces Vanishing Saliency of Recurrent Neural Networks

**Aya Abdelsalam Ismail[1], Mohamed Gunady[1], Luiz Pessoa[2]**
**Héctor Corrada Bravo[*1] , Soheil Feizi [*1]**
{asalam,mgunady}@cs.umd.edu, pessoa@umd.edu,
hcorrada@umiacs.umd.edu, sfeizi@cs.umd.edu
[1] Department of Computer Science, University of Maryland
[2] Department of Psychology, University of Maryland

## Abstract

Recent efforts to improve the interpretability of deep neural networks use saliency to characterize the importance of input features to predictions made by models. Work on interpretability using saliency-based methods on Recurrent Neural Networks (RNNs) has mostly targeted language tasks, and their applicability to time series data is less understood. In this work we analyze saliency-based methods for RNNs, both classical and gated cell architectures. We show that RNN saliency vanishes over time, biasing detection of salient features only to later time steps and are, therefore, incapable of reliably detecting important features at arbitrary time intervals. To address this vanishing saliency problem, we propose a novel RNN cell structure (input-cell attention[†]), which can extend any RNN cell architecture. At each time step, instead of only looking at the current input vector, input-cell attention uses a fixed-size matrix embedding, each row of the matrix attending to different inputs from current or previous time steps. Using synthetic data, we show that the saliency map produced by the input-cell attention RNN is able to faithfully detect important features regardless of their occurrence in time. We also apply the input-cell attention RNN on a neuroscience task analyzing functional Magnetic Resonance Imaging (fMRI) data for human subjects performing a variety of tasks. In this case, we use saliency to characterize brain regions (input features) for which activity is important to distinguish between tasks. We show that standard RNN architectures are only capable of detecting important brain regions in the last few time steps of the fMRI data, while the input-cell attention model is able to detect important brain region activity across time without latter time step biases.

## 1 Introduction

Deep Neural Networks (DNNs) are successfully applied to a variety of tasks in different domains, often achieving accuracy that was not possible with conventional statistical and analysis methods. Nevertheless, practitioners in fields such as neuroscience, medicine, and finance are hesitant to use models like DNNs that can be difficult to interpret. For example, in clinical research, one might like to ask, "Why did you predict this person as more likely to develop Alzheimer's disease?" Making DNNs amenable to queries like this remains an open area of research.

The problem of interpretability for deep networks has been tackled in a variety of ways [20, 30, 22, 3, 19, 13, 14, 16, 18, 21, 31]. The majority of this work focuses on vision and language tasks and their application to time series data, specifically when using recurrent neural nets (RNNs), is poorly

---

[*]Authors contributed equally
[†]Code available at https://github.com/ayaabdelsalam91/Input-Cell-Attention

understood. Interpretability in time series data requires methods that are able to capture changes in the importance of features over time. The goal of our paper is to understand the applicability of feature importance methods to time series data, where detecting importance in specific time intervals is necessary. We will concentrate on the use of saliency as a measure of feature importance.

As an illustration of the type of problem we seek to solve, consider the following task classification problem from neuroimaging [25]: a subject is performing a certain task (e.g., a memory or other cognitive task) while being scanned in an fMRI machine. After preprocessing of the raw image signal, the data will consist of a multivariate time series, with each feature measuring activity in a specific brain region. To characterize brain region activity pertinent to the task performed by the subject, a saliency method should be able to capture changes in feature importance (corresponding to brain regions) over time. This is in contrast to similar text classification problems [2], where the goal of saliency methods is to give a relevance score to each word in the sequence, whereas the saliency of individual features in each word embedding is not important.

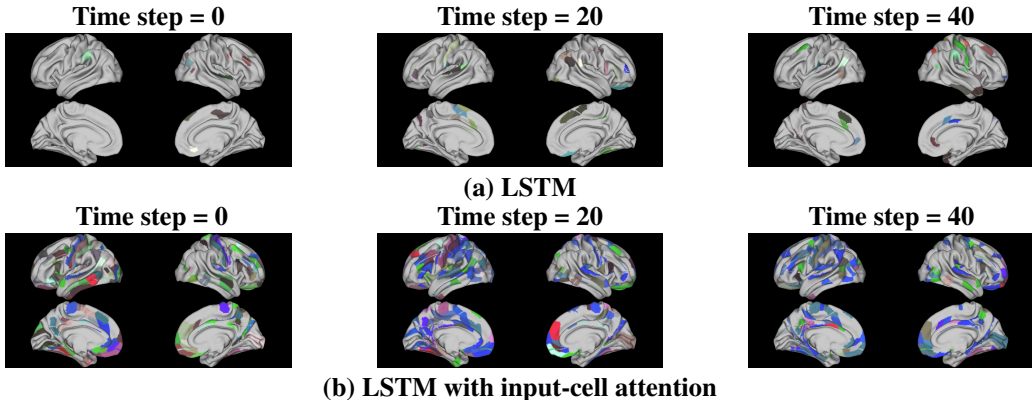

Figure 1: A subject performs a task while scanned by an fMRI machine. Images are processed and represented as a multivariate time series, with each feature corresponding to a brain region. RNNs are used to classify time series based on the task performed by the subject. Figure (a) shows the saliency map produced by LSTM. Importance detected at later time steps (40) is significantly higher then that detected in earlier time steps. Figure (b) shows the saliency map produced by LSTM with input-cell attention. We observe no time interval bias in the detected importance.

Motivated by problems of this type, our paper presents three main contributions:

1. We study the effect of RNNs, specifically LSTMs, on saliency for time series data and show theoretically and empirically that saliency vanishes over time and is therefore incapable of reliably detecting important features at arbitrary time intervals on RNNs.

2. We propose and evaluate a modification for LSTMs ("input-cell attention") that applies an attention mechanism to the input of an LSTM cell (Figure 2) allowing the LSTM to "attend" to time steps that it finds important.

3. We apply input-cell attention to an openly available fMRI dataset from the Human Connectome Project (HCP) [26], in a task classification setting and show that by using input-cell attention we are able to capture changes in the importance of brain activity across time in different brain regions as subjects perform a variety of tasks (Figure 1).

"Gating mechanisms", introduced in LSTMs [9], help RNN models carry information from previous time steps, thus diminishing the vanishing gradient problem to improve prediction accuracy. We show, however, that these mechanisms do not diminish the vanishing gradient problem enough to allow saliency to capture feature importance at arbitrary time intervals. For example, Figure 1a shows the saliency map produced by an LSTM applied to the task classification problem outlined above. In this case, the LSTM reports feature importance only in the last few time steps, ignoring the earlier ones.

The input-cell attention mechanism uses a fixed-size matrix embedding at each time step $t$, to represent the input sequence up to time $t$. Each row of the embedding matrix is designed to attend

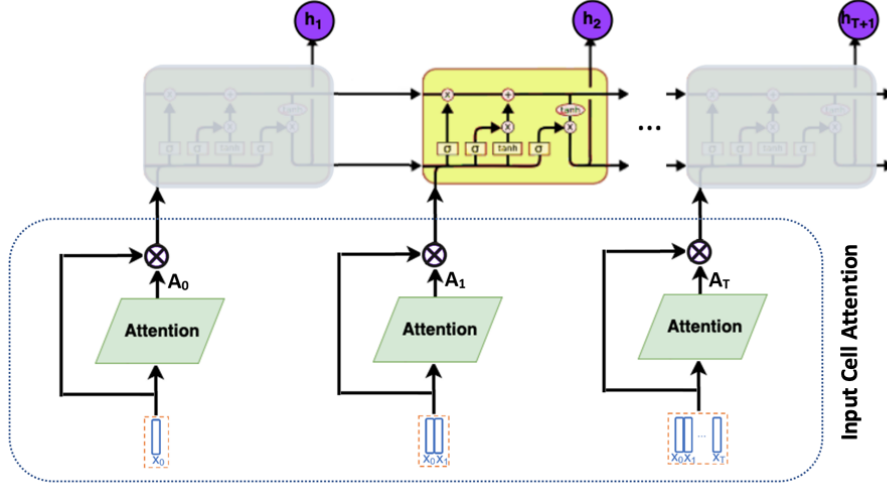

Figure 2: LSTM with input-cell attention, at time $t$ matrix $X_t = [x_0, x_1, \ldots, x_t]$ is passed to an attention mechanism; the output $A_t$ is multiplied with $X_t$ to produce $M_t$ (i.e $M_t = A_t X_t$). Matrix $M_t$ is now the input to LSTM cell ($M_t$ has dimension $r \times N$, where $r$ is the attention parameter and $N$ is the number of inputs).

to different inputs including time $t$ or previous time steps. This provides a direct gradient path from the output at the final time step, through the embedding matrix, to all input time steps thereby circumventing the vanishing saliency problem (Figure 1b). We show via simulation and application to our illustrative neuroimaging task classification problem, that saliency maps produced by input-cell attention are able to faithfully detect important features regardless of their occurrence in time.

## 2 Related Work

**Attribution methods** include perturbation-based methods [30] that compute the attribution of an input feature by measuring the difference in network's output with and without that feature. Other methods compute the attributions for all input features by backpropagating through the network [19, 24, 3, 20] this is known as gradient-based or backpropagation attribution methods. It has been proven by Ancona et al. [1] that complex gradient-based attribution methods including $\epsilon$-LRP [3] and DeepLift [19] can be reformulated as computing backpropagation for a modified gradient function. Since the goal of our paper is to study the behavior of feature importance detection in RNNs, we have chosen saliency, perhaps the simplest gradient-based attribution method, to represent the other, more complex, gradient-based attribution methods.

**Neural attention mechanisms** are popular techniques that allow models to attend to different input features of interest. Bahdanau et al. [4] used attention for alignment in machine translation. Xu et al. [28] implemented attention for computer vision to identify important regions of an image. In addition, attention was also used to extract important portions of text in a document [29, 7]. Lin et al. [12] deployed self-attention to create a sentence embedding by attending to the hidden state of each word in the sentence. Vaswani et al. [27] introduced the Transformer, a neural architecture based solely on attention mechanisms. Current attention mechanisms are mainly applied to hidden states across time steps. In contrast, we utilize attention in this work to detect salient features over time without bias towards the last time steps, by attending on different time steps of an input at the cell level of RNNs.

**Feature visualization** is an attempt to better understand LSTMs by visualizaiton of hidden state dynamics. Hasani et al. [6] ranks the contribution of individual cells to the final output to help understand LSTM hidden state dynamics. LSTMVis [23] explains individual cell's functionality by matching local hidden-state patterns to similar ones in larger networks. IMV-LSTM [5] uses a mixture attention mechanism to summarize contribution of specific features on hidden state. Karpathy et al. [11] uses character-level language models as an interpretable testbed. Olah et al. [15] presents general user visual interfaces to explore model interpretation measures from DNNs.

# 3 Problem Definition

We study the problem of assigning feature importance, or "relevance", at a given time step to each input feature in a network. We denote input as $X = (x_1, x_2, \ldots, x_t, \ldots, x_T)$, where $T$ is the last time step and vector $x_t = [x_{t_1}, \ldots, x_{t_N}] \in \mathbb{R}^N$ is the feature vector at time step $t$. $N$ is the number of features, $x_{t_i}$ is input feature $i$ at time $t$. An RNN takes input $X$ and produces output $S(X) = [S_1(X), \ldots, S_C(X)]$, where $C$ is the total number of output neurons. Given a specific target neuron $c$, the goal is to find the contribution $R^c = [R_{1_1}^c, \ldots, R_{t_1}^c, \ldots, R_{t_N}^c, \ldots, R_{T_N}^c] \in R^N$ of each input feature $x_{t_i}$ to the output $S_c$.

# 4 Vanishing Saliency: a Recurrent Neural Network Limitation

Consider the example shown in figure (3a), where all important features are located in the first few time steps (red box) and the rest is Gaussian noise. One would expect the saliency map to highlight the important features at the beginning of the time series. However, the saliency produced by LSTM (Figure 3b) shows some feature importance at the last few time steps with no evidence of importance at the earlier ones. Methods such as hidden layer pooling and self-attention [12] are used to consider outputs from different time steps but they fail in producing a reasonable saliency map (refer to section 6.1 and supplementary material for more details). In this section we investigate the reasons behind LSTM's bias towards last few time steps in saliency maps.

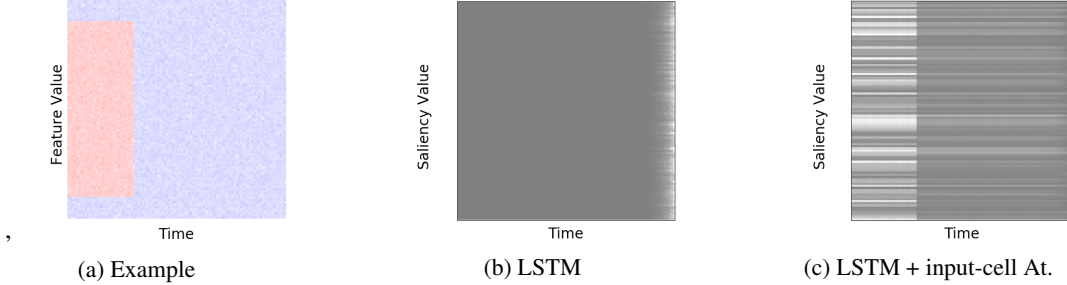

,

    (a) Example                       (b) LSTM                  (c) LSTM + input-cell At.

Figure 3: (a) A sample from a simulated dataset where the horizontal axis represents time and vertical axis represents feature values. (b) Saliency map produced by LSTM; importance is only captured in the last few time steps. (c) Saliency map produced by LSTM with input-cell attention; our model is able to differentiate between important and non-important feature regardless of there location in time.

The gating mechanisms of LSTM [9] are shown in equation (1), where $\sigma(\cdot)$ denotes the sigmoid (i.e. logistic) function and $\odot$ denotes element-wise vector product. LSTM has three gates: input , forget and output gates, given as $i_t$, $f_t$ and $o_t$ respectively. These gates determine whether or not to let new input in ($i_t$), delete information from all previous time steps ($f_t$) or to let it impact the output at the current time step ($o_t$).

$$
\begin{aligned}
\mathbf{i}_t &= \sigma\left(\mathbf{W}_{xi}\mathbf{x}_t + \mathbf{W}_{hi}\mathbf{h}_{t-1} + \mathbf{b}_i\right) \\
\mathbf{f}_t &= \sigma\left(\mathbf{W}_{xf}\mathbf{x}_t + \mathbf{W}_{hf}\mathbf{h}_{t-1} + \mathbf{b}_f\right) \\
\mathbf{o}_t &= \sigma\left(\mathbf{W}_{xo}\mathbf{x}_t + \mathbf{W}_{ho}\mathbf{h}_{t-1} + \mathbf{b}_o\right) \\
\tilde{\mathbf{c}}_{\mathbf{t}} &= \tanh\left(\mathbf{W}_{x\tilde{c}}\mathbf{x}_t + \mathbf{W}_{h\tilde{c}}\mathbf{h}_{t-1} + \mathbf{b}_{\tilde{c}}\right) \\
\mathbf{c}_t &= \mathbf{f}_t \odot \mathbf{c}_{t-1} + \mathbf{i}_t \odot \tilde{\mathbf{c}}_{\mathbf{t}} \\
\mathbf{h}_t &= \mathbf{o}_t \odot \tanh\left(\mathbf{c}_t\right)
\end{aligned}
\tag{1}
$$

The amount of saliency preserved is controlled by $f_t$; this can be demonstrated by calculating the saliency $R_T^c(x_t)$ where $t < T$ (further details in supplementary material)

$$
\begin{aligned}
R_T^c(x_t) &= \left| \frac{\partial S_c(x_T)}{\partial x_t} \right| \\
&= \left| \frac{\partial S_c}{\partial h_T} \left( \prod_{i=T}^{t+1} \frac{\partial h_i}{\partial h_{i-1}} \right) \frac{\partial h_t}{\partial x_t} \right|
\end{aligned}
$$

$\frac{\partial h_t}{\partial h_{t-1}}$ is the only term affected by the number of time steps. Solving its partial derivative we get,

$$\frac{\partial h_t}{\partial h_{t-1}} = \tanh\left(c_t\right)\left(\boxed{W_{ho}}\left(o_t \odot \left(1 - o_t\right)\right)\right) + o_t \odot \left(1 - \tanh^2\left(c_t\right)\right)\left[c_{t-1}\left(\boxed{W_{hf}}\left(f_t \odot \left(1 - f_t\right)\right)\right)+\right.$$
$$\tilde{c}_t\left(\boxed{W_{hi}}\left(i_t \odot \left(1 - i_t\right)\right)\right)+$$
$$\left.i_t\left(\boxed{W_{h\tilde{c}}}\left(1 - \tilde{c}_t \odot \tilde{c}_t\right)\right) + f_t\right]$$

As $t$ decreases (i.e earlier time steps), those terms multiplied by the weight matrix (black box in above equation) will eventually vanish if the largest eigenvalue of the weight matrix is less then 1, this is known as the "vanishing gradient problem" [8]. $\frac{\partial h_t}{\partial h_{t-1}}$ will be reduced to :

$$\frac{\partial h_t}{\partial h_{t-1}} \approx o_t \odot \left(1 - \tanh^2\left(c_t\right)\right)\left[f_t\right]$$

From the equation above, one can see that the amount of information preserved depends on the LSTM's "forget gate" ($f_t$); hence, as $t$ decreases (i.e earlier time steps) its contribution to the relevance decreases and eventually disappears, as we empirically observe in figure (3b).

## 5    Input-Cell Attention For Recurrent Neural Networks

To address the vanishing saliency problem described in Section 4, we propose a novel RNN cell structure, called "input-cell attention." The proposed cell structure is shown in figure (2); at each time step $t$, instead of looking only at the current input vector $x_t$, all inputs accumulated and available to current time steps are considered by passing them through an attention mechanism. The attention module provides a set of summation weight matrices for the inputs. The set of summation weight vectors is multiplied with the inputs, producing a fixed size matrix of weighted inputs $M_t$ attending to different time steps. $M_t$ is then passed to the LSTM cell. To accommodate the changes in gates inputs, the classical LSTM gating equations are changed from those shown in (1) to the new ones shown (2). Note that input-cell attention can be added to any RNN cell architecture; however, the LSTM architecture is the focus of this paper.

$$\begin{aligned}
\mathbf{i}_t &= \sigma\left(\boxed{\mathbf{W}_{Mi}\mathbf{M}_t} + \mathbf{W}_{hi}\mathbf{h}_{t-1} + \mathbf{b}_i\right) \\
\mathbf{f}_t &= \sigma\left(\boxed{\mathbf{W}_{Mf}\mathbf{M}_t} + \mathbf{W}_{hf}\mathbf{h}_{t-1} + \mathbf{b}_f\right) \\
\mathbf{o}_t &= \sigma\left(\boxed{\mathbf{W}_{Mo}\mathbf{M}_t} + \mathbf{W}_{ho}\mathbf{h}_{t-1} + \mathbf{b}_o\right) \\
\tilde{\mathbf{c}}_\mathbf{t} &= \tanh\left(\boxed{\mathbf{W}_{M\tilde{c}}\mathbf{M}_t} + \mathbf{W}_{h\tilde{c}}\mathbf{h}_{t-1} + \mathbf{b}_{\tilde{c}}\right) \\
\mathbf{c}_t &= \mathbf{f}_t \odot \mathbf{c}_{t-1} + \mathbf{i}_t \odot \tilde{\mathbf{c}}_\mathbf{t} \\
\mathbf{h}_t &= \mathbf{o}_t \odot \tanh\left(\mathbf{c}_t\right)
\end{aligned} \tag{2}$$

We use the same attention mechanism that was introduced for self-attention [12]. However, in our architecture attention is performed at the cell level rather than the hidden layer level. Using the same notation as in section 3, matrix $X_t = [x_1, \ldots, x_t]$ with dimensions $t \times N$ where $N$ is the size of the feature embedding. The attention mechanism takes $X_t$ as input, and outputs a matrix of weights $A_t$:

$$A_t = \text{softmax}\left(W_2 \tanh\left(W_1 X_t^T\right)\right) \tag{3}$$

$W_1$ is a weight matrix with dimensions $d_a \times N$ where $d_a$ is a hyper-parameter. The number of time steps the attention mechanism will attend to is $r$, known as "attention hops". $W_2$ is also a weight matrix that has dimension $r \times d_a$. Finally, the output weight matrix $A_t$ has dimension $r \times t$; $A_t$ has a weight for each time step and the $\text{softmax}()$ ensures that all the computed weights sum to 1. The inputs $X_t$ are projected linearly according to the weights provided by $A_t$ to get a matrix $M_t = A_t X_t$ with fixed dimension $r \times N$. One can the view attention mechanism as a two-layer

unbiased feed-forward network, with parameters $\{W_1, W_2\}$ and $d_a$ hidden units. $M_t$ is flattened to a vector of length $r * N$ and passed as the input to the LSTM cell as shown in Figure 2. The dimensions of each learned weight matrix $W_x$ in the standard LSTM equation (1) is $N \times h$, where $h$ is size of hidden layer. The input-cell attention weight matrix $W_M$ the learned parameters in equation (2) have dimensions $h \times (r * N)$.

**Approximation:** To reduce the dimensionality of the LSTM input at each time step, matrix $M_t$ can be modified to be the average of features across attention hops. By doing so, the value of a feature in the embedding will equal its average value across all time steps the model attends to. As mentioned previously, $M_t$ has dimensions $r \times N$ let $m_{ij}$ be value of feature $j$ at attention hop $i$

$$\widetilde{m_j} = \frac{\sum_{i=1}^{r} m_{ij}}{r} \tag{4}$$

This reduces matrix $M_t$ to vector $\widetilde{m_t}$ with dimension $N$. The dimensions of weight matrix $W_M$ equations (2) return to $N \times h$ as in original LSTM equations (1). Self-attention [12] used this approximation in the github code they provided. We used this version of input-cell attention for the experiments in Section (6).

## 6    Experimental Results

### 6.1    Synthetic Data for Evaluation

To capture the behavior of saliency methods applied to RNNs, we create synthetic labeled datasets with two classes. A single example is Gaussian noise with zero mean and unit variance; a box of 1's is added to the important features in the positive class or subtracted in the negative class; the feature embedding size for each sample $N = 100$ and the number of time steps $T = 100$. This configuration helps us differentiate between feature importance in different time intervals. The specific features and the time intervals (boxes) on which they are considered important is varied between datasets to test each model's ability to capture importance at different time intervals. Figure 4 shows 3 example datasets. The same figure also shows how important time intervals and features are specified in various experimental setups.

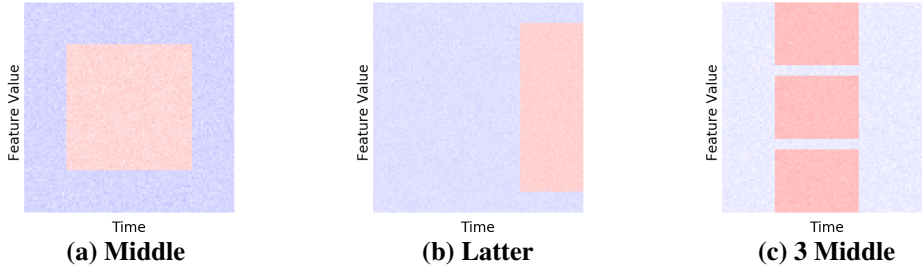

Figure 4: Synthetic Datasets, where red represents important features and blue is Gaussian noise.

#### 6.1.1    Saliency Performance Measurements

**Euclidean distance:** Since we know the time interval of important features in each example, we create a reference sample which has value 1 for important features and 0 for noise. We measure the normalized Euclidean distance between the saliency map $R(X)$ produced by each model for given sample $X$ (where $X = [x_1, \ldots, x_n]$) and its reference sample ref, the distance is calculated by the equation below, where $n = N \times T$

$$\frac{\sum_{i=1}^{n} \sqrt{(\text{ref}_i - R(x_i))^2}}{\sum_{i=1}^{n} \text{ref}_i} \tag{5}$$

**Weighted Jaccard similarity [10]:** The value of saliency represents the importance of the feature at a specific time. We measure the concordance between the set of high saliency features to the known

set of important features in simulation. Jaccard measures similarity as the size of the intersection divided by the size of the union of the sample sets, meaning that high values and low ones have equal weight. Weighted Jaccard addresses this by considering values, since the higher saliency value represents higher importance, it is a better measure of similarity for this problem. Weighted Jaccard similarity $J$ between absolute value of sample $|X|$ and its saliency $R(X)$ is defined as

$$J\left(|X|, R(X)\right) = \frac{\sum_{i=1}^{n} \min\left(|x_i|, R(x_i)\right)}{\sum_{i=1}^{n} \max\left(|x_i|, R(x_i)\right)} \tag{6}$$

### 6.1.2 Performance on Synthetic Datasets

We compared LSTMs with input-cell attention with standard LSTMs [9], bidirectional LSTMs [17] and LSTMs with self-attention [12] (other LSTMs with various pooling architectures were also compared; performance is reported in the supplementary material).

**Static Box Experiments:** To test how the methods perform when important features are located at different time steps we create: "Earlier Box" dataset figure (3a), "Middle Box" dataset figure (4a) and "Latter Box" datasets figure (4b); important features are located from $t_0$ to $t_{30}$, from $t_{30}$ to $t_{70}$ and from $t_{70}$ to $t_{100}$ respectively; the results are shown in the table (1a).

To avoid bias we also tested on 1. "Mixed Boxes" dataset in which the location of the importance box differs in each sample. 2. "3 Earlier Boxes", "3 Latter Boxes" and "3 Middle Boxes"(similar to figure 4c ) where not all features are important at one specific time.; the results are shown in the table (1b).

LSTM with input-cell attention outperforms other methods in both metrics for all datasets. One important observation is that LSTM performance is higher in the latter box problem, which aligns with our observed bias towards reporting importance in later time steps.

| Model | Ealier Box | | | Middle Box | | | Latter Box | | |
|---|---|---|---|---|---|---|---|---|---|
| | WJac | Euc | Acc | WJac | Euc | Acc | WJac | Euc | Acc |
| LSTM | 0.000 | 1.006 | 53.4 | 0.000 | 1.003 | 98.6 | 0.019 | 0.985 | 100.0 |
| Bi-LSTM | 0.000 | 1.004 | 50.7 | 0.000 | 1.003 | 53.2 | 0.013 | 0.990 | 100.0 |
| LSTM+self At. | 0.048 | 0.973 | 100.0 | 0.048 | 0.963 | 100.0 | 0.045 | 0.973 | 100.0 |
| LSTM+in.cell At. | **0.103** | **0.914** | 100.0 | **0.124** | **0.891** | 100.0 | **0.110** | **0.903** | 100.0 |

(a)

| Model | Mixed Boxes | | | 3 Ealier Boxes | | | 3 Middle Boxes | | |
|---|---|---|---|---|---|---|---|---|---|
| | WJac | Euc | Acc | WJac | Euc | Acc | WJac | Euc | Acc |
| LSTM | 0.000 | 1.003 | 49.1 | 0.000 | 1.003 | 52.2 | 0.000 | 1.004 | 51.8 |
| Bi-LSTM | 0.000 | 1.002 | 51.5 | 0.000 | 1.003 | 51.3 | 0.000 | 1.003 | 51.3 |
| LSTM+self At. | 0.060 | 0.953 | 100.0 | 0.025 | 0.985 | 100.0 | 0.075 | 0.939 | 99.9 |
| LSTM+in.cell At. | **0.104** | **0.912** | 77.6 | **0.108** | **0.903** | 100.0 | **0.106** | **0.905** | 100.0 |

(b)

Table 1: Saliency performance: weighted Jaccard (WJac) and Euclidean distance (Euc). For LSTM, bidirectional LSTM, LSTM with self-attention and LSTM with input-cell attention on different datasets where important features are located at different time steps (ACC is the model accuracy).

**Moving Box Experiments:** To identify the time step effect on the presence of a feature in a saliency map, we created five datasets that differ in the start and end time of importance box; images from the datasets are available in the supplementary material. This experiment reports the effect of changing the feature importance time interval on the ability of each method to detect saliency.

We plot the change of weighted Jaccard and Euclidean distance against the change in starting time step of important features in Figure 5. LSTM with input-cell attention and LSTM with self-attention are unbiased towards time. However, LSTM with input-cell attention outperforms LSTM with self-attention in both metrics. Classical LSTM and bidirectional LSTM are able to detect salient features at later time steps only (comparison with other architectures is in the supplementary material).

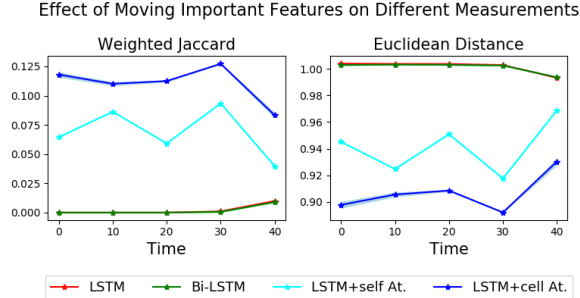

Figure 5: The effect of changing the location of importance features in time on weighted Jaccard (WJac) and Euclidean distance (Euc). For LSTM, bidirectional LSTM, LSTM with self-attention and LSTM with input-cell attention.

## 6.2 MNIST as a Time Series

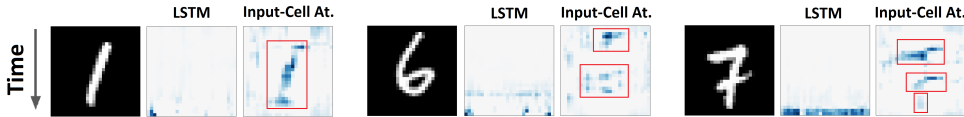

Figure 6: Saliency maps of samples from MNIST with time as y-axis. Saliency maps are shown for both vanilla LSTM and LSTM with input-cell attention. The vanishing gradients in the saliency is clear in LSTM which fails to provide informative maps, whereas adding input-cell attention recovers gradient values for features at different time steps.

In the previous synthetic datasets, we evaluated saliency maps obtained by different approaches on a simple setting were continuous blocks of important features are distributed over time. In order to validate the resulting saliency maps in cases where important features have more structured distributions of different shapes, we treat the MNIST image dataset as a time series. In other words, a $28 \times 28$ image is turned into a sequence of 28 time steps, each of which is a vector of 28 features. Time is represented in the y-axis. For more interpretable visualization of saliency maps, we trained the models to perform a three-class classification task by subsetting the dataset to learn only the digits "1", "6", and "7". These digits were selected since the three share some common features, while having distinctive features at different time steps.

Both standard LSTMs and LSTMs with input-cell attention were trained to convergence. Figure 6 shows the saliency maps for three samples; saliency maps obtained from LSTMs exhibit consistent decay over time. When assisted with input-cell attention mechanism, our architecture overcomes that decay and can successfully highlight important features at different time steps. Supplementary material shows more samples exhibiting similar behavior.

## 6.3 Human Connectome Project fMRI Data

To evaluate our method in a more realistic setting, we apply input-cell attention to an openly available fMRI dataset of the Human Connectome Project (HCP) [26]. In this dataset, subjects are performing certain tasks while scanned by an fMRI machine. Our classification problem is to identify the task performed given the fMRI scans (more details about tasks and preprocessing fMRI data is available in supplementary material). Recurrent networks have been used for this task before, e.g. in the DLight framework [25]. DLight uses LSTM to analyze whole-brain neuro-imaging data then applies layer-wise relevance propagation (LRP) [3, 2] to the trained model identifying those regions of interest in the brain (ROIs) whose activity the model used to make a prediction. However, this framework gives a single interpretation for the entire time series; applying input-cell attention enables us to see changes in the importance of brain region activity across time.

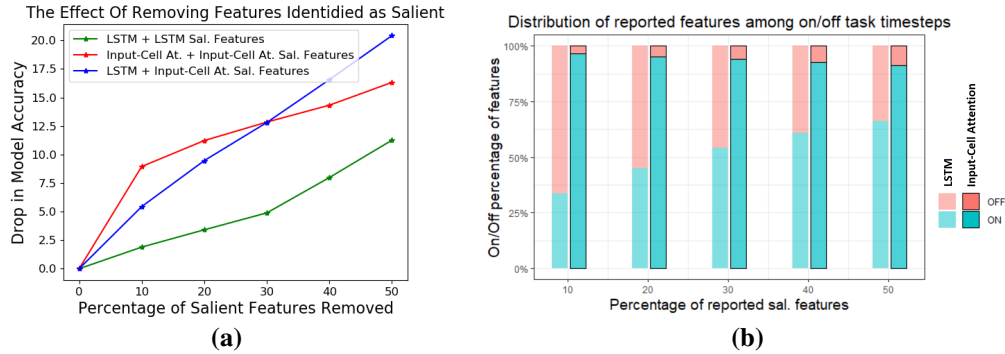

Figure 7: (a) The effect of dropping salient features identified by each model. Dropping LSTM's top 10% salient features reduces accuracy by 2%, while for LSTM with input-cell attention accuracy dropped by 9.5%. If features identified as salient by LSTM with input-cell attention are removed from standard LSTM its accuracy drops 6%. (b) Percentage of on-task off-task features identified as salient, more than 70% of top 10% salient features identified by LSTM are from off-task period.

### 6.3.1 Experiments and Results

We performed two types of experiments: **(1) On-Task:** data was taken while the subject was actively performing task. **(2) On-Task off-Task:** data was taken while the subject was both actively performing task (on-task) and during rest period between tasks (off-task). The off-task time is used as a negative control for importance at the end of the time series since models should not be able to differentiate between tasks based on data obtained during off-task periods. More details about the experimental setup can be found in the supplementary material.

**On-Task Experiment:** First we trained an LSTM on a binary classification task until convergence. On each correctly classified task we produced a saliency map. We plotted the saliency map to see which ROIs are important while the subject is performing the task. Figure (1a) shows that the LSTM was only able to capture changes in ROI importance at the last few time steps. We repeated the same experiment using LSTM with input-cell attention with results shown in figure (1b). Input-cell attention was able to capture changes in importance for different brain regions across time that were not recovered by LSTM.

**On-Task Off-Task Experiment:** Models were first trained on the off-task period only, accuracy produced by the models was random confirming our assumption that off-task period data does not contain any useful information for task classification. Models were then trained on the on-task period followed by off-task period, saliency maps were used to identify important features. Figure (7 a) shows the effect of removing features identified as salient on model accuracy (note that the model with the ability of correctly detect salient features will result in a larger drop in accuracy on feature removal). Figure (7 b) shows percentage of on-task off-task features identified as salient by each model. Our architecture is faithfully able to detect salient features during on-task portions of time.

## 7 Summary and Conclusion

We have shown empirically and theoretically that saliency maps produced by LSTMs vanish over time. Importance is only ascribed to later time steps in a time series and earlier time steps are not considered. We reduced this vanishing saliency problem by applying an attention mechanism at the cell level. By attending to inputs across different time steps, the LSTM was able to consider important features from previous time steps. We applied our methods to fMRI data from the Human Connectome Project and observed the same phenomenon of LSTM vanishing saliency in a task detection problem. This last result, taken together with a belief that assigning importance only to the last few time steps in this neuro-imaging application severely limits the interpretability of LSTM models, and considering our results on synthetic data, indicates that our work opens a path towards solving a critical shortcoming in the application of modern recurrent DNNs to problems where interpretability of time series models is important.

## Acknowledgments

This work was supported by the U.S. National Institutes of Health grant [R01GM 114267] and NSF award [CDS&E:1854532]. The funders had no role in study design, data collection and analysis, decision to publish, or preparation of the manuscript.

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
