[Supplementary Material]

# Supplementary Materials: Input-Cell Attention Reduces Vanishing Saliency of Recurrent Neural Networks

## Vanishing Saliency: a Recurrent Neural Network Limitation

Calculating the saliency $R_T^c(x_T)$ for feature embedding $x$ at last time step $T$ given output $c$ is fairly simple,

$$R_T^c(x_T) = \left| \frac{\partial S_c(x_T)}{\partial x_T} \right|$$

$$\frac{\partial S_c(x_T)}{\partial x_T} = \frac{\partial S_c}{\partial h_T} \frac{\partial h_T}{\partial x_T}$$

The value of $x_T$ directly contributes to $S_c(x_T)$; hence, $R_T^c(x_T)$ is relatively high. Now let's consider saliency for $x_t$ where $t < T$.

$$R_t^c(x_t) = \left| \frac{\partial S_c(x_T)}{\partial x_t} \right|$$

$$\frac{\partial S_c(x_T)}{\partial x_t} = \frac{\partial S_c}{\partial h_T} \frac{\partial h_T}{\partial h_t} \frac{\partial h_t}{\partial x_t}$$

$$\frac{\partial S_c(x_T)}{\partial x_t} = \frac{\partial S_c}{\partial h_T} \left( \prod_{i=T}^{t+1} \frac{\partial h_i}{\partial h_{i-1}} \right) \frac{\partial h_t}{\partial x_t}$$

$\frac{\partial h_i}{\partial h_{i-1}}$ is the only term affected by the number of time steps; we can expand it as:

$$\frac{\partial h_t}{\partial h_{t-1}} = \frac{\partial h_t}{\partial o_t} \frac{\partial o_t}{\partial h_{t-1}} + \frac{\partial h_t}{\partial c_t} \frac{\partial c_t}{\partial h_{t-1}}$$

$$= \frac{\partial h_t}{\partial o_t} \frac{\partial o_t}{\partial h_{t-1}} + \frac{\partial h_t}{\partial c_t} \left[ \frac{\partial c_t}{\partial f_t} \frac{\partial f_t}{\partial h_{t-1}} + \frac{\partial c_t}{\partial i_t} \frac{\partial h_{t-1}}{\partial c_t} + \frac{\partial c_t}{\partial \tilde{c}_t} \frac{\partial \tilde{c}_t}{\partial h_{t-1}} + \frac{\partial c_t}{\partial c_{t-1}} \right]$$

Plugging the partial derivative in the above formula, we get:

$$\frac{\partial h_t}{\partial h_{t-1}} = \tanh(c_t) \left( \boxed{W_{ho}} \left( o_t \odot (1 - o_t) \right) \right) + o_t \odot \left( 1 - \tanh^2(c_t) \right) \left[ c_{t-1} \left( \boxed{W_{hf}} \left( f_t \odot (1 - f_t) \right) \right) + \right.$$
$$\tilde{c}_t \left( \boxed{W_{hi}} \left( i_t \odot (1 - i_t) \right) \right) +$$
$$\left. i_t \left( \boxed{W_{h\tilde{c}}} \left( 1 - \tilde{c}_t \odot \tilde{c}_t \right) \right) + f_t \right]$$

As $t$ decreases (i.e earlier time steps), those terms multiplied by the weight matrix (black box in above equation) will eventually vanish if the largest eigenvalue of the weight matrix is less then 1; this is known as the "vanishing gradient problem". $\frac{\partial h_t}{\partial h_{t-1}}$ will be reduced to :

$$\frac{\partial h_t}{\partial h_{t-1}} \approx o_t \odot \left( 1 - \tanh^2(c_t) \right) \left[ f_t \right]$$

From the equation above, one can see that the amount of information preserved depends on the LSTM's "forget gate" ($f_t$); hence, as $t$ decreases (i.e earlier time steps) its contribution to the relevance decreases and eventually disappears as we empirically observe.

Figure 1: Example of synthetic datasets that are used throughout the paper, where red represents important features and blue is Gaussian noise

## Synthetic Data experiments

**Static Box Experiments:**

| Model | Ealier Box | | | Middle Box | | | Latter Box | | |
|---|---|---|---|---|---|---|---|---|---|
| | WJac | Euc | Acc | WJac | Euc | Acc | WJac | Euc | Acc |
| LSTM | 0.000 | 1.006 | 53.4 | 0.000 | 1.003 | 98.6 | 0.019 | 0.985 | 100.0 |
| Bi-LSTM | 0.000 | 1.004 | 50.7 | 0.000 | 1.003 | 53.2 | 0.013 | 0.990 | 100.0 |
| LSTM+in.cell | 0.103 | 0.914 | 100.0 | **0.124** | **0.891** | 100.0 | **0.110** | **0.903** | 100.0 |
| LSTM+Max pl | 0.002 | 1.006 | 99.9 | 0.001 | 1.004 | 100.0 | 0.002 | 1.006 | 100.0 |
| LSTM+Max pl+in.cell | 0.076 | 0.931 | 100.0 | 0.015 | 0.990 | 99.8 | 0.011 | 1.002 | 100.0 |
| LSTM+Mean pl | 0.007 | 1.024 | 99.9 | 0.038 | 0.974 | 100.0 | 0.033 | 0.997 | 99.9 |
| LSTM+Mean pl+in.cell | 100.0 | 0.904 | 100.0 | 0.029 | 0.982 | 99.9 | 0.003 | 1.010 | 98.0 |
| LSTM+self At. | 0.048 | 0.973 | 100.0 | 0.048 | 0.963 | 100.0 | 0.045 | 0.973 | 100.0 |
| LSTM+self At.+in.cell | **0.124** | **0.878** | 100.0 | 0.014 | 0.994 | 99.9 | 0.014 | 0.995 | 100.0 |

(a)

| Model | Mixed Boxes | | | 3 Ealier Boxes | | | 3 Middle Boxes | | |
|---|---|---|---|---|---|---|---|---|---|
| | WJac | Euc | Acc | WJac | Euc | Acc | WJac | Euc | Acc |
| LSTM | 0.000 | 1.003 | 49.1 | 0.000 | 1.003 | 52.2 | 0.000 | 1.004 | 51.8 |
| Bi-LSTM | 0.000 | 1.002 | 51.5 | 0.000 | 1.003 | 51.3 | 0.000 | 1.003 | 51.3 |
| LSTM+in.cell | **0.104** | **0.912** | 77.6 | 0.108 | 0.903 | 100.0 | **0.106** | **0.905** | 100.0 |
| LSTM+Max pl | 0.003 | 1.003 | 100.0 | 0.002 | 1.003 | 100.0 | 0.002 | 1.004 | 99.9 |
| LSTM+Max pl+in.cell | 0.009 | 0.997 | 100.0 | 0.053 | 0.953 | 100.0 | 0.012 | 0.993 | 99.9 |
| LSTM+Mean pl | 0.034 | 0.979 | 100.0 | 0.014 | 1.005 | 100.0 | 0.067 | 0.946 | 100.0 |
| LSTM+Mean pl+in.cell | 0.033 | 0.977 | 100.0 | **0.124** | **0.879** | 100.0 | 0.012 | 0.995 | 100.0 |
| LSTM+self At. | 0.060 | 0.953 | 100.0 | 0.025 | 0.985 | 100.0 | 0.075 | 0.939 | 99.9 |
| LSTM+self At.+in.cell | 0.014 | 0.992 | 100.0 | 0.091 | 0.916 | 100.0 | 0.043 | 0.967 | 100.0 |

(b)

Table 1: Saliency performance: weighted Jaccard (WJac) and Euclidean distance (Euc). For the following architectures: (1) LSTM (2) bidirectional LSTM (3) LSTM with input-cell attention (4) LSTM with Max pooling (5) LSTM with Max pooling and input-cell attention (6) LSTM with Mean pooling (7) LSTM with Mean pooling and input-cell attention (8) LSTM with self-attention (9) LSTM with self-attention and input-cell attention on different datasets where important features are located at different time steps (ACC is the model accuracy on test data).

**Moving Box Experiments:**

(a) Datasets with different start and end time for important features

(b) The effect of changing the location of importance features in time on weighted Jaccard (WJac) and Euclidean distance (Euc) for different models.

**Partial Attention Experiments:**

The following experiment we study the effect of having partial input-cell attention (input-cell attention is only applied to some time steps). Figure 3 shows an experiment where we applied attention only to the last 10 time steps for middle box dataset (Figure 1b) . The Figure illustrates that attention on the last few time steps in the partial attention case helped preserve saliency longer then that of vanilla LSTM; however, saliency eventually vanishes in both cases. To preserve importance through time, at each time step model needs to attend to different inputs from current or previous time steps.

(a) LSTM                (b) LSTM + partial input-cell At.            (c) LSTM + input-cell At.

Figure 3: This Figure shows saliency map from different models on a sample from middle box simulated dataset. (a) Saliency map produced by LSTM; importance is only captured in the last few time steps. (a) Saliency map produced by LSTM with input-cell attention applied to the last 10 time steps only; importance is captured longer then LSTM however it eventually vanishes. (c) Saliency map produced by LSTM with input-cell attention; our architecture is able to differentiate between important and non-important feature regardless of there location in time.

## MNIST Dataset

Here we present more results for random samples from MNIST dataset.

Figure 4: Saliency maps for more samples from MNIST on 3 digits "1", "6", "7" with time as y-axis. Heatmaps are shown for both Vanilla LSTM and LSTM with input-cell attention. Important features are present at different time steps for each class ("7" important features are in early time steps, whereas they are in middle/late time steps for "6"). The vanishing gradients in the saliency is clear in LSTM, whereas adding input-cell attention recovers gradient values for features in early time steps.

## Human Connectome Project fMRI Data

### Dataset Description:

We used three tasks in HCP data:

- **Gambling:** Participants play a card guessing game where they are asked to guess the number on a mystery card (represented by a ?) in order to win or lose money.

- **Relational Processing:** Participants are presented with 2 pairs of objects, with one pair at the top of the screen and the other pair at the bottom of the screen. They are told that they should first decide what dimension differs across the top pair of objects (differed in shape or differed in texture) and then they should decide whether the bottom pair of objects also differ along that same dimension.

- **Working Memory:** Participants were presented pictures of places, tools, faces and body parts we refer to pictures as stimulus. Participants performed a "2-back" working memory task, where they indicated if the current stimulus matched the one presented two stimuli before, or a control condition called "0-back" (without a memory component).

HCP provides a minimally prepossessed released dataset; in addition to their preprocessing, we regressed out 12 motion-related variables using the 3dDeconvolve routine of the AFNI package [1] and low frequency signal. We only considered cortical data, then we employed the cortical parcellation developed by the HCP research group [2]. The parcellation produced 360 cortical regions of interest (ROIs); meaning at each time step we have a feature vector of size 360, representing various brain regions.

### Experiment Setup:

All experiments were performed using data from 566 subjects for training and 183 for testing. HCP data is divided into blocks, there are two types of blocks (a) active blocks: where subjects were actively performing the task. (b) non-active blocks: subjects are resting this includes task cues and time between different runs.

### On-Task Experiment:

This is a binary classification task between gambling and relational processing, only active blocks were considered the length of time series is around 43 time steps.

**On-Task Off-Task Experiment:**

This is a binary classification task between gambling and working memory these tasks were chosen because they have similar active block length and both contain equal cue time. Each sample contains 38 time steps from an active block followed by 10 time steps from a non-active block.

**Additional results from On-Task Off-Task Experiment:**

Figure 5: Distribution of salient features reported by: LSTM and LSTM+ input-cell attention over different time steps. The on-task time window of 40 frames are divided into 4 buckets, followed by an off-task bucket where subjects were annotated non active.