[Reviews · NeurIPS 2019]

Reviewer 1



The paper starts by showing empirically and theoretically that saliency maps generated using gradient vanishes over long sequences in LSTMs. The authors propose a modification to the LSTM cell, called LSTM with cell-attention, which can attend to all previous time steps. They show that this approach improve considerably the saliency on the input sequence. They also test their approach on the fMRI dataset of the Human Connectome Project (HCP). Originality: The proposed LSTM with cell-attention is a novel combination of well-known techniques. The proposed synthetic datasets seem to be original and will probably be used in the future in other works. While the paper focus more on the use of RNNs for time series modeling, the authors do not provide any literature review of other approaches on time series such as convolutional neural networks as well as other machine learning models. Quality: - In the theoretical part of the paper, from the laid out gradient equation of the LSTM cell (after line 97), it is not clear to me what terms eventually vanish and why. A more thorough explanation of this would be appreciated. However, using the synthetic datasets, the authors show well that the gradient does not flow to every important time step when using the standard LSTM cell. With the same datasets, they also show that the proposed approach is able to overcome this problem. - It is not very clear to me how the saliency performance measures were chosen or designed. In the case of the Euclidean distance, it seems that the value of 1 in the reference sample is quite arbitrary. If the saliency value R(x_i) goes over 1, it incurs a penalty whereas it should not since it greatly indicates that the feature x_i is important. In the case of the weighted Jaccard similarity, it is not clear what is the intuition behind the similarity between the example and its saliency map. Furthermore, the weighted Jaccard similarity was designed for non-negative values. However, the features of the synthetic datasets can be negative. For both performance measures, the authors should explain the intuition behind them. - Another minor comment: it is to be noted that the synthetic datasets take for granted that important events in the time series extend on several time steps instead single events separated by multiple time steps. This might be worth exploring in future work. - Other than a simple figure, results on the fMRI dataset were not discussed in depth nor compared with any other works. - Choi et al.[1] did similar work on time series. The authors should compare their work with this. Clarity: - As a general note, I would like to say that I had to read multiple times several parts of the paper to be able to understand correctly what the authors meant. These parts were generally simple concepts explained in a convoluted way. - In particular, these sentences after line 118 are expressed poorly: “The number of time steps the attention mechanism will attend to is r, known as ‘attention hops’. Note that the larger r is, the more time steps will be considered. E.g., if r = t, then we are assigning weights for all previous and current time steps; whereas with r = 1, only a single time step is considered.” In current works on attention mechanisms, r could be considered as the number of attention heads. It does not really restrict the number of time steps considered. The authors meant that having multiple attention heads can help the network to focus on multiple time steps since one head is sometimes able to only look at a single time step. However, a head can often look at more than one single time step. - As a note on the vocabulary used in the paper, the definition of self-attention of Lin et al. (2017)[2] is not up-to-date with current research. Particularly, the definition of self-attention by Vaswani et al. (2017)[3] is currently the most used. It can be quite confusing for the reader to use this old definition. - There are a couple of places where line numbers are missing. Also, most equations do not have a number. These two facts render review of the paper a bit harder. - Few other formatting comments: Typo on Line 27: DNNs (lowercase s) Caption of Figure 1: time series (instead of timeseries) Typo on Line 99: i instead of t Paragraph below line 137: Unneeded space before the paragraph Section 5.1.1 Euclidean distance, reformulate as: the saliency map R(X) produced by each model for given sample X Lines 148 to 154: Description of Figure 4 not accurate and missing letters d and e. Figure 1 in supplementary material: colors are inverted by comparison to Figure 4 in paper. Typo on Line 155: cell-attention outperforms Significance: - As said previously, I am sure that the synthetic datasets will be used in future publications to address similar problems in time series. - The proposed LSTM with cell-attention is no more than an attention mechanism followed by an LSTM cell. The improved saliency of the approach comes from the attention done in the last time steps of the RNN. Even though saliency on important features is improved, the approach ignores most of the temporal nature of the problem. Thus, the proposed approach does not solve the saliency vanishing problem of LSTMs. - The experiment on the fMRI dataset do not really show anything more than what previously laid out with the synthetic datasets. [1] Choi, E., Bahadori, M. T., Sun, J., Kulas, J., Schuetz, A., & Stewart, W. (2016). Retain: An interpretable predictive model for healthcare using reverse time attention mechanism. In Advances in Neural Information Processing Systems (pp. 3504-3512). [2] Lin, Z., Feng, M., Santos, C. N. D., Yu, M., Xiang, B., Zhou, B., & Bengio, Y. (2017). A structured self-attentive sentence embedding. arXiv preprint arXiv:1703.03130. [3] Vaswani, A., Shazeer, N., Parmar, N., Uszkoreit, J., Jones, L., Gomez, A. N., ... & Polosukhin, I. (2017). Attention is all you need. In Advances in neural information processing systems (pp. 5998-6008). Post rebuttal response ----------------------------- - Essentially, the authors do not modify the LSTM cell in any way, they only add attention over their inputs. The LSTM weights themselves then still suffers from the highlighted vanishing gradient problem and do not learn anything specific to the first time steps. Thus, the paper does not propose an approach that "Reduces Vanishing Saliency of Recurrent Neural Networks" as it claims to. In their response, the authors proposed an experiment with partial cell attention. From their response, it is unclear to me what it really consists of. A possible explanation of the obtained result is that the input distribution (i.e. from raw input to attended input) given to the LSTM has changed in the last time steps and the LSTM may have reacted badly to this change of distribution. In any case, I am really perplex of this result and need more convincing facts to support it (other than the authors "did it wrong"). They could have used something like RETAIN [1] to test the hypothesis that "attention is all you need". Or simply a simple attention attention mechanism followed by fully-connected layers and residual connections. - For the fMRI dataset, one would have expected a more detailed analysis of the results when reading the paper. - I like the proposed synthetic datasets. While the authors did not do a good job at explaining the intuition behind their metrics, I finally did understand from their response. For the contribution of the synthetic datasets and metrics, I am willing to give a score of 5 since I think these really are the main contributions of the paper.

Reviewer 2



Originality: This work builds a novel solution to an existing challenge of RNNs based on already known techniques with the goal of improving interpretability and expressivity of the machine learning model. The differences to the previous work have been clearly stated, but a related work section is missing. Quality: The theoretical contributions of the paper are technically sound. The experimental part can be improved, though. Although the toy examples are very clear and advocates for the soundness of the results. More experimental evaluations are needed. Moreover, a large body of related works are missing in the paper; specifically, fundamental attempts on the interpretability of deep networks and specifically recurrent neural networks. Examples include: Attribution (Saliency maps) [Olah et al., Distill 2018]. Dimensionality reduction [Hinton et al. JMLR 2008] Feature visualization [Karpathy et al., ICLR 2016] Pattern matching [Strobelts, et al., IEEE Trans Computer Vision and Graphics 2018] Attention Networks [Bengio et al. ICLR 2015] Dynamical systems analysis of LSTM nets [Hasani et al. ArXiv 2018] Clarity: The paper is well-written and well-organized. The structure of the paper makes it easy to follow the paper’s storyline. A related work section is missing, and this makes the judgment of the clarity of the results a bit challenging. Significance: This work takes a step forward towards addressing an issue regarding the interpretability of recurrent cells, the “vanishing saliency problem”. The theoretical description of the problem is novel and technically sound. I find the investigation of the paper over the “vanishing saliency problem” quite interesting. I have a suggestion/concern, if we have the vanishing saliency problem in RNNs, and the solution is to use an additional attention mechanism at the input of the LSTM networks, what would happen if we do not use an LSTM in this case (both the toy and the real-life examples) and only use an Attention network? I would suggest to include this experiment and provide results. Maybe really “attention is all you need”? Although it is invaluable to improve performance over a real-life test case, in my opinion, the experimentations performed in the paper could be extended to at least one of the standard benchmarks to compare performance. It is beneficial to observe the attention weight matrices, where one could reason about how the dependencies to the initial time steps are being established. Also, it would be interesting to see how the learned forget-gate weights differ in networks of Vanilla LSTM compared to LSTM networks supplied with an attention mechanism.

Reviewer 3



While many attention applications are focused on weighted averages of vector states from previous time steps as an input to an LSTM, this paper is interested n attending not just to the sequence steps but individual features in the input. The goal is to obtain an interpretable model using saliency as a model introspection mechanism. While in this settings traditional saliency vanishes beyond a few timesteps in the past, the proposed solution carrying all timesteps with each input to LSTM is able to provide the interpretability across the input sequence. It is unclear from the paper whether the saliency is computed for each timestep that the LSTM made, or rather for each input accumulated and available to the current time step. I suspect the latter, but correct me if I am wrong. If I am right though, then it makes complete sense why the proposed trick would work better than any attempts for "flowing" the gradients to the past through all LSTM positions. However, the proposed model regardless of the saliency is now more of accumulative rather than sequential, where both the cell and the weighted input gradually accumulate the sequence. Because of that difference in the model semantics, the approach may not be widely generalizable, although it looks like the problem of highlighting feature importance across time can be addressed in this fashion. It is unclear why the paper insists on calling their input level attention (A_t) attention at the _cell level_ Clearly, equation 2 shows accumulation of the attention weighted input, not the attention weighted cell elements. Probably, just a confusing naming. Please, consider changing. 1. Please cite the Attention paper by Bahdanau, Cho, Bengio ’14 2. It is unclear what happens in the paragraph between lines 118 and 119. The unnumbered expression for A_t does not make it clear whether the matrix that's passed as an argument to softmax is treated as a flattened vector and all the elements of the resulting A_t sum to 1 or only those for each feature within a time-step? 3. Lines 120 and 121 are unclear as well. W_{M} is called a matrix but it is a tensor of rxNxh dimension. Does this mean that M is flattened to a vector of length r*N and W_{M} is indeed a matrix of h x (r*N) dimension?

[Author Response · NeurIPS 2019]

**Response to Reviewer 1:**

**"It is not clear what terms eventually vanish and why."** Terms multiplied by weight matrix $W$ (red boxes) in

equation after line 97 vanish if the largest eigenvalue of matrix $W$ is less then 1. Multiplying it repeatedly causes the

vanishing gradient. This reduces equation after line 97 to equation in line 100.

**"Euclidean distance is penalized if saliency goes over 1."** We use the normalized Euclidean distance as mentioned

in line 157 so the distance will never go over 1.

**"The fMRI dataset do not show anything more than what previously laid out"** The goal of the dataset is to show

a real-case application where we are interested in seeing how features importance change across time. Previous work

[1] that has been done on this dataset gives a single interpretation for the entire time series and does not show how

features change across time. To clarify our point we will add the off-task time to our experiments (time where subject is

listening to instructions not preforming the actual task) and we will investigate the ability of our model to ignore this

time and put the importance on the on-task time.

**"The definition of self-attention is not up-to-date."** We choose to use the definition of self-attention of Lin et al.

(2017) since they are applying self-attention to a RNN, similar to our case. Also, we do not use the same attention

function described by Vaswani et al. (2017).

**"Improved saliency comes from the attention done in the last time steps of the RNN."** We disagree. If this was

true then removing attention from first time steps would make no difference which is not the case. Figure 1a shows an

experiment where we applied attention only to the last 10 time steps (referred to as partial cell attention) for middle

box dataset, saliency still vanishes. Our proposed method improves saliency because, at each time step, cell-attention

attends to different inputs from current or previous time steps preserving importance through time.

**"Proposed approach ignores the temporal nature of the problem".** We disagree. Our entire paper is based on time

series, permuting data in time and producing different saliency is the entire purpose of our synthetic dataset. The

moving box experiment in line 158 and figure 1 in supplementary material shows a clear example where the only

difference between samples is their location in time, it is very clear from the experiments that we do *NOT* ignore the

temporal nature of data.

**Minor comments.** For weighted Jaccard, we compare the saliency with the absolute value of synthetic data sample, we

will update this in the final version. We will add numbering to all equations, correct mentioned typos and fix coloring

for figure 1 in supplementary material.

**Response to Reviewer 2:** Thank you for your comments. In the original version of the paper, we mentioned related

work briefly in the introduction but we did not have an entire section dedicated to related work due to space limitation.

In the revised draft, we will add a related work section and make sure we cite and explain all papers you listed in this

section along with others. Our scope in this paper is on studying saliency of RNNs where we propose an approach to

resolve the vanishing saliency that hinders the interpretation of such networks. Thus, we have kept the comparison with

vanilla attention and other non-recurrent network architectures to our future work. Thank you for suggesting to add

other standard benchmarks to our experiments. Upon your suggestion, we decided to add two new benchmarks. (1) We

use MNIST dataset as a time series data where one dimension of the 2D images acts as the time axis (a $28 \times 28$ image

is turned into a sequence of 28 time steps, each of which is a vector of 28 features). We choose MNIST because it offers

an interpretable visualization. Figure 1b is an example of a saliency map produced for vanilla LSTM and our proposed

LSTM + cell attention. (2) CMU Multimodal Opinion Sentiment Intensity (MOSI, Zadeh et. al 2016) , a dataset of

opinion level sentiment intensity in online videos. In the final version, we will include these new experimental results.

Finally, we will include distributions of weight matrices of the network, as suggested.

**Response to Reviewer 3:** Thank you for your comments. You are correct that the saliency is computed for each input

captured and accumulated till the current time step. We will make sure to make this point more clear in the final

manuscript. The accumulation effect is reduced by the approximation mentioned in the paragraph under line 121. We

called our method cell attention because its attention is on the cell level rather than hidden layer level although we

understand your concern about this name and how this might create some confusion. We may consider changing the

name of the method to Recurrent Attention. For lines 118 and 119: $A_t$ has dimensions $r \times t$ where $t$ is the number of

time steps in the current input, $A_t$ has a weight for each time step; weight of all time steps sum up to 1. For lines 120

and 121 $M$ is flattened to a vector of length $r * N$ and $W_M$ is a matrix of $h \times (r * N)$. Thank you for pointing this out

we will make sure this is corrected in our final version.

(a) Experiment showing that having cell-attention **only** in the last time steps (Partial Cell Atten.) still produces vanishing saliency.

(b) An example of saliency map produced for MNIST, when treated as a time series. Saliency vanishes for vanilla LSTM while our proposed model is able to detect important features.

[1] Thomas, Armin W., et al. "Interpretable LSTMs for whole-brain neuroimaging analyses."


[Meta-Review · NeurIPS 2019]

Although the writing style of the paper could be improved and the presence of some perplexity about the experimental setting, the paper contributes in a valuable way to the advancement of the area. Moreover, the rebuttal helped to clarify most of the issues raised by the most negative reviewer. Overall the merits of the paper seem to overcome its drawbacks.